# GENERATIVE ADVERSARIAL NETWORK TRAINING IS A CONTINUAL LEARNING PROBLEM

## ABSTRACT

Generative Adversarial Networks (GANs) have proven to be a powerful framework for learning to draw samples from complex distributions. However, GANs are also notoriously difficult to train, with mode collapse and oscillations a common problem. We hypothesize that this is at least in part due to the evolution of the generator distribution and the catastrophic forgetting tendency of neural networks, which leads to the discriminator losing the ability to remember synthesized samples from previous instantiations of the generator. Recognizing this, our contributions are twofold. First, we show that GAN training makes for a more interesting and realistic benchmark for continual learning methods evaluation than some of the more canonical datasets. Second, we propose leveraging continual learning techniques to augment the discriminator, preserving its ability to recognize previous generator samples. We show that the resulting methods add only a light amount of computation, involve minimal changes to the model, and result in better overall performance on the examined image and text generation tasks.

## 1 INTRODUCTION

Generative Adversarial Networks (Goodfellow et al., 2014) (GANs) are a popular framework for modeling draws from complex distributions, demonstrating success in a wide variety of settings, for example image synthesis (Radford et al., 2016; Karras et al., 2018) and language modeling (Li et al., 2017). In the GAN setup, two agents, the *discriminator* and the *generator* (each usually a neural network), are pitted against each other. The generator learns a mapping from an easy-to-sample latent space to a distribution in the data space, which ideally matches the real data's distribution. At the same time, the discriminator aims to distinguish the generator's synthesized samples from the real data samples. When trained successfully, GANs yield impressive results; in the image domain for example, synthesized images from GAN models are significantly sharper and more realistic than those of other classes of models (Larsen et al., 2016). On the other hand, GAN training can be notoriously finicky. One particularly well-known and common failure mode is *mode collapse* (Che et al., 2017; Srivastava et al., 2017): instead of producing samples sufficiently representing the true data distribution, the generator maps the entire latent space to a limited subset of the real data space.

When mode collapse occurs, the generator does not "converge," in the conventional sense, to a stationary distribution. Rather, because the discriminator can easily learn to recognize a mode-collapsed set of samples and the generator is optimized to avoid the discriminator's detection, the two end up playing a never-ending game of cat and mouse: the generator meanders towards regions in the data space the discriminator thinks are real (likely near where the real data lie) while the discriminator chases after it. Interestingly though, if generated samples are plotted through time (as in Figure 1), it appears that the generator can revisit previously collapsed modes. At first, this may seem odd. The discriminator was ostensibly trained to recognize that mode in a previous iteration and did so well enough to push the generator away from generating those samples. Why has the discriminator seemingly lost this ability?

We conjecture that this oscillation phenomenon is enabled by *catastrophic forgetting* (McCloskey & Cohen, 1989; Ratcliff, 1990): neural networks have a well-known tendency to forget how to complete old tasks while learning new ones. In most GAN models, the discriminator is a binary classifier, with the two classes being the real data and the generator's outputs. Implicit to the training of a standard classifier is the assumption that the data are drawn independently and identically distributed (i.i.d.). Importantly, this assumption does *not* hold true in GANs: the distribution of the generator

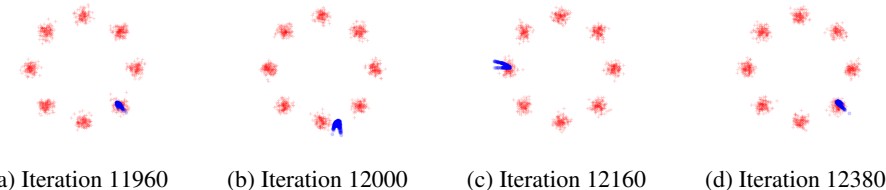

(a) Iteration 11960      (b) Iteration 12000      (c) Iteration 12160      (d) Iteration 12380

Figure 1: Real samples from a mixture of eight Gaussians in red; generated samples in blue. (a) The generator is mode-collapsed in the bottom right. (b) The discriminator learns to recognize the generator oversampling this region and pushes the generator away, so the generator gravitates toward a new mode. (c) The discriminator continues to chase the generator, causing the generator to move in a clockwise direction. (d) The generator eventually returns to the same mode as (a). Such oscillations are common while training a vanilla GAN. Best seen as a video: https://youtu.be/91a2gPWngo8.

class (and thus the discriminator's training data) evolves over time. Moreover, these changes in the generator's distribution are adversarial, designed specifically to deteriorate discriminator performance on the fake class as much as possible. Thus, the alternating training procedure of GANs in actuality corresponds to the discriminator learning tasks sequentially, where each task corresponds to recognizing samples from the generator at that particular point in time. Without any measures to prevent catastrophic forgetting, the discriminator's ability to recognize fake samples from previous iterations will be clobbered by subsequent gradient updates, allowing a mode-collapsed generator to revisit old modes if training runs long enough. Given this tendency, a collapsed generator can wander indefinitely without ever learning the true distribution.

With this perspective in mind, we cast training the GAN discriminator as a continual learning problem, leading to two main contributions. (*i*) While developing systems that learn tasks in a sequential manner without suffering from catastrophic forgetting has become a popular direction of research, current benchmarks have recently come under scrutiny as being unrepresentative to the fundamental challenges of continual learning (Farquhar & Gal, 2018). We argue that GAN training is a more realistic setting, and one that current methods tend to fail on. (*ii*) Such a reframing of the GAN problem allows us to leverage relevant methods to better match the dynamics of training the min-max objective. In particular, we build upon the recently proposed *elastic weight consolidation* (Kirkpatrick et al., 2017) and *intelligent synapses* (Zenke et al., 2017). By preserving the discriminator's ability to identify previous generator samples, this memory prevents the generator from simply revisiting past distributions. Adapting the GAN training procedure to account for catastrophic forgetting provides an improvement in GAN performance for little computational cost and without the need to train additional networks. Experiments on CelebA and CIFAR10 image generation and COCO Captions text generation show discriminator continual learning leads to better generations.

## 2   BACKGROUND: CATASTROPHIC FORGETTING IN GANS

Consider distribution $p_{\mathrm{real}}(\boldsymbol{x})$, from which we have data samples $\mathcal{D}^{\mathrm{real}}$. Seeking a mechanism to draw samples from this distribution, we learn a mapping from an easy-to-sample latent distribution $p(\boldsymbol{z})$ to a data distribution $p_{\mathrm{gen}}(\boldsymbol{x})$, which we want to match $p_{\mathrm{real}}(\boldsymbol{x})$. This mapping is parameterized as a neural network $G_{\boldsymbol{\phi}}(\boldsymbol{z})$ with parameters $\boldsymbol{\phi}$, termed the generator. The synthesized data are drawn $\boldsymbol{x} = G_{\boldsymbol{\phi}}(\boldsymbol{z})$, with $\boldsymbol{z} \sim p(\boldsymbol{z})$. The form of $p_{\mathrm{gen}}(\boldsymbol{x})$ is not explicitly assumed or learned; rather, we learn to draw samples from $p_{\mathrm{gen}}(\boldsymbol{x})$.

To provide feedback to $G_{\boldsymbol{\phi}}(\boldsymbol{z})$, we simultaneously learn a binary classifier that aims to distinguish synthesized samples $\mathcal{D}^{\mathrm{gen}}$ drawn from $p_{\mathrm{gen}}(\boldsymbol{x})$ from the true samples $\mathcal{D}^{\mathrm{real}}$. We also parameterize this classifier as a neural network $D_{\boldsymbol{\theta}}(\boldsymbol{x}) \in [0, 1]$ with parameters $\boldsymbol{\theta}$, with $D_{\boldsymbol{\theta}}(\boldsymbol{x})$ termed the discriminator. By incentivizing the generator to fool the discriminator into thinking its generations are actually from the true data, we hope to learn $G_{\boldsymbol{\phi}}(\boldsymbol{z})$ such that $p_{\mathrm{gen}}(\boldsymbol{x})$ approaches $p_{\mathrm{real}}(\boldsymbol{x})$.

These two opposing goals for the generator and discriminator are usually formulated as the following min-max objective:

$$\min_{\boldsymbol{\phi}} \max_{\boldsymbol{\theta}} \mathcal{L}^{\mathrm{GAN}}(\boldsymbol{\theta}, \boldsymbol{\phi}) = \mathbb{E}_{\boldsymbol{x} \sim p_{\mathrm{real}}(\boldsymbol{x})}[\log D_{\boldsymbol{\theta}}(\boldsymbol{x})] + \mathbb{E}_{\boldsymbol{z} \sim p(\boldsymbol{z})}[\log(1 - D_{\boldsymbol{\theta}}(G_{\boldsymbol{\phi}}(\boldsymbol{z})))] \qquad (1)$$

At each iteration $t$, we sample from $p_{\text{gen}}(\boldsymbol{x})$, yielding generated data $\mathcal{D}_t^{\text{gen}}$. These generated samples, along with samples from $\mathcal{D}^{\text{real}}$, are then passed to the discriminator. A gradient descent optimizer nudges $\boldsymbol{\theta}$ so that the discriminator takes a step towards maximizing $\mathcal{L}^{\text{GAN}}(\boldsymbol{\theta}, \boldsymbol{\phi})$. Parameters $\boldsymbol{\phi}$ are updated similarly, but to minimize $\mathcal{L}^{\text{GAN}}(\boldsymbol{\theta}, \boldsymbol{\phi})$. These updates to $\boldsymbol{\theta}$ and $\boldsymbol{\phi}$ take place in an alternating fashion. The expectations are approximated using samples from the respective distributions, and therefore learning only requires observed samples $\mathcal{D}^{\text{real}}$ and samples from $p_{\text{gen}}(\boldsymbol{x})$.

The updates to $G_{\boldsymbol{\phi}}(\boldsymbol{z})$ mean that $p_{\text{gen}}(\boldsymbol{x})$ changes as a function of $t$, perhaps substantially. Consequently, samples $\{\mathcal{D}_1^{\text{gen}}, ..., \mathcal{D}_t^{\text{gen}}\}$ come from a sequence of different distributions. At iteration $t$, only samples from $\mathcal{D}_t^{\text{gen}}$ are available, as $G_{\boldsymbol{\phi}}(\boldsymbol{z})$ has changed, and saving previous instantiations of the generator or samples $\{\mathcal{D}_1^{\text{gen}}, ..., \mathcal{D}_{t-1}^{\text{gen}}\}$ can be prohibitive. Thus, $D_{\boldsymbol{\theta}}(\boldsymbol{x})$ is typically only provided $\mathcal{D}_t^{\text{gen}}$, so it only learns the most recent distribution, with complete disregard for previous $p_{\text{gen}}(\boldsymbol{x})$. Because of the catastrophic forgetting effect of neural networks, the ability of $D_{\boldsymbol{\theta}}(\boldsymbol{x})$ to recognize these previous distributions is eventually lost in the pursuit of maximizing $\mathcal{L}^{\text{GAN}}(\boldsymbol{\theta}, \boldsymbol{\phi})$ with respect to *only* $\mathcal{D}_t^{\text{gen}}$. This opens the possibility that the generator goes back to generating samples the discriminator had previously learned (and then forgot) to recognize, leading to unstable mode-collapsed oscillations that hamper GAN training (as in Figure 1). Recognizing this problem, we propose that the discriminator should be trained with the temporal component of $p_{\text{gen}}(\boldsymbol{x})$ in mind.

## 3 METHOD

### 3.1 CLASSIC CONTINUAL LEARNING

Catastrophic forgetting has long been known to be a problem with neural networks trained on a series of tasks (McCloskey & Cohen, 1989; Ratcliff, 1990). While there are many approaches to addressing catastrophic forgetting, here we primarily focus on elastic weight consolidation (EWC) and intelligent synapses (IS). These are meant to illustrate the potential of catastrophic forgetting mitigation to improve GAN learning, with the expectation that this opens up the possibility of other such methods to significantly improve GAN training, at low additional computational cost.

#### 3.1.1 ELASTIC WEIGHT CONSOLIDATION (EWC)

To derive the EWC loss, Kirkpatrick et al. (2017) frames training a model as finding the most probable values of the parameters $\boldsymbol{\theta}$ given the data $\mathcal{D}$. For two tasks, the data are assumed partitioned into independent sets according to the task, and the posterior for Task 1 is approximated as a Gaussian with mean centered on the optimal parameters for Task 1 $\boldsymbol{\theta}_1^*$ and diagonal precision given by the diagonal of the Fisher information matrix $F_1$ at $\boldsymbol{\theta}_1^*$. This gives the EWC loss the following form:

$$\mathcal{L}(\boldsymbol{\theta}) = \mathcal{L}_2(\boldsymbol{\theta}) + \mathcal{L}^{\text{EWC}}(\boldsymbol{\theta}), \quad \text{with} \quad \mathcal{L}^{\text{EWC}}(\boldsymbol{\theta}) \triangleq \frac{\lambda}{2} \sum_i F_{1,i}(\theta_i - \theta_{1,i}^*)^2 , \qquad (2)$$

where $\mathcal{L}_2(\boldsymbol{\theta}) = \log p(\mathcal{D}_2|\boldsymbol{\theta})$ is the loss for Task 2 individually, $\lambda$ is a hyperparameter representing the importance of Task 1 relative to Task 2, $F_{1,i} = \left(\frac{\partial \mathcal{L}_1(\boldsymbol{\theta})}{\partial \theta_i}\big|_{\boldsymbol{\theta}=\boldsymbol{\theta}_1^*}\right)^2$, $i$ is the parameter index, and $\mathcal{L}(\boldsymbol{\theta})$ is the new loss to optimize while learning Task 2. Intuitively, the EWC loss prevents the model from straying too far away from the parameters important for Task 1 while leaving less crucial parameters free to model Task 2. Subsequent tasks result in additional $\mathcal{L}^{\text{EWC}}(\boldsymbol{\theta})$ terms added to the loss for each previous task. By protecting the parameters deemed important for prior tasks, EWC as a regularization term allows a single neural network (assuming sufficient parameters and capacity) to learn new tasks in a sequential fashion, without forgetting how to perform previous tasks.

#### 3.1.2 INTELLIGENT SYNAPSES (IS)

While EWC makes a point estimate of how essential each parameter is at the conclusion of a task, IS (Zenke et al., 2017) protects the parameters according to their importance along the task's entire training trajectory. Termed synapses, each parameter $\theta_i$ of the neural network is awarded an *importance measure* $\omega_{1,i}$ based on how much it reduced the loss while learning Task 1. Given a loss gradient $\boldsymbol{g}(t) = \nabla_{\boldsymbol{\theta}} \mathcal{L}(\boldsymbol{\theta})|_{\boldsymbol{\theta}=\boldsymbol{\theta}_t}$ at time $t$, the total change in loss during the training of Task 1 then is the sum of differential changes in loss over the training trajectory. With the assumption that parameters $\boldsymbol{\theta}$ are independent, we have:

$$\int_{t^0}^{t^1} \boldsymbol{g}(t)d\boldsymbol{\theta} = \int_{t^0}^{t^1} \boldsymbol{g}(t)\boldsymbol{\theta}'dt = \sum_i \int_{t^0}^{t^1} g_i(t)\theta_i'dt \triangleq -\sum_i \omega_{1,i} , \qquad (3)$$

where $\boldsymbol{\theta}' = \frac{d\boldsymbol{\theta}}{dt}$ and $(t^0, t^1)$ are the start and finish of Task 1, respectively. Note the added negative sign, as importance is associated with parameters that decrease the loss.

The importance measure $\omega_{1,i}$ can now be used to introduce a regularization term that protects parameters important for Task 1 from large parameter updates, just as the Fisher information matrix diagonal terms $F_{1,i}$ were used in EWC. This results in an IS loss very reminiscent in form[1]:

$$\mathcal{L}(\boldsymbol{\theta}) = \mathcal{L}_2(\boldsymbol{\theta}) + \mathcal{L}^{\text{IS}}(\boldsymbol{\theta}), \quad \text{with} \quad \mathcal{L}^{\text{IS}}(\boldsymbol{\theta}) \triangleq \frac{\lambda}{2} \sum_i \omega_{1,i}(\theta_i - \theta_{1,i}^*)^2 . \tag{4}$$

## 3.2 GAN CONTINUAL LEARNING

The traditional continual learning methods are designed for certain canonical benchmarks, commonly consisting of a small number of clearly defined tasks (e.g., classification datasets in sequence). In GANs, the discriminator is trained on dataset $\mathcal{D}_t = \{\mathcal{D}^{\text{real}}, \mathcal{D}_t^{\text{gen}}\}$ at each iteration $t$. However, because of the evolution of the generator, the distribution $p_{\text{gen}}(x)$ from which $\mathcal{D}_t^{\text{gen}}$ comes changes over time. This violates the i.i.d. assumption of the order in which we present the discriminator data. As such, we argue that different instances in time of the generator should be viewed as separate tasks. Specifically, in the parlance of continual learning, the training data are to be regarded as $\mathcal{D} = \{(\mathcal{D}^{\text{real}}, \mathcal{D}_1^{\text{gen}}), (\mathcal{D}^{\text{real}}, \mathcal{D}_2^{\text{gen}}), ...\}$. Thus motivated, we would like to apply continual learning methods to the discriminator, but doing so is not straightforward for the following reasons:

- **Definition of a task**: EWC and IS were originally proposed for discrete, well-defined tasks. For example, Kirkpatrick et al. (2017) applied EWC to a DQN (Mnih et al., 2015) learning to play ten Atari games sequentially, with each game being a clear, independent task. For GAN, there is no such precise definition as to what constitutes a "task," and as discriminators are not typically trained to convergence at every iteration, it is also unclear how long a task should be.

- **Computational memory**: While Equations 2 and 4 are for two tasks, they can be extended to $K$ tasks by adding a term $\mathcal{L}^{\text{EWC}}$ or $\mathcal{L}^{\text{IS}}$ for each of the $K - 1$ prior tasks. As each term $\mathcal{L}^{\text{EWC}}$ or $\mathcal{L}^{\text{IS}}$ requires saving both a historical reference term $\boldsymbol{\theta}_k^*$ and either $F_k$ or $\boldsymbol{\omega}_k$ (all of which are the same size as the model parameters $\boldsymbol{\theta}$) for each task $k$, employing these techniques naively quickly becomes impractical for bigger models when $K$ gets large, especially if $K$ is set to the number of training iterations $T$.

- **Continual *not* learning**: Early iterations of the discriminator are likely to be non-optimal, and without a forgetting mechanism, EWC and IS may forever lock the discriminator to a poor initialization. Additionally, the unconstrained addition of a large number of terms $\mathcal{L}^{\text{EWC}}$ or $\mathcal{L}^{\text{IS}}$ will cause the continual learning regularization term to grow unbounded, which can disincentivize any further changes in $\boldsymbol{\theta}$.

To address these issues, we build upon the aforementioned continual learning techniques, and propose several changes.

**Number of tasks as a rate**: We choose the total number of tasks $K$ as a function of a constant rate $\alpha$, which denotes the number of iterations before the conclusion of a task, as opposed to arbitrarily dividing the GAN training iterations into some set number of segments. Given $T$ training iterations, this means a rate $\alpha$ yields $K = \frac{T}{\alpha}$ tasks.

**Online Memory**: Seeking a way to avoid storing extra $\boldsymbol{\theta}_k^*$, $F_k$, or $\boldsymbol{\omega}_k$, we observe that the sum of two or more quadratic forms is another quadratic, which gives the classifier loss with continual learning the following form for the $(k + 1)^{\text{th}}$ task:

$$\mathcal{L}(\boldsymbol{\theta}) = \mathcal{L}_{k+1}(\boldsymbol{\theta}) + \mathcal{L}^{\text{CL}}(\boldsymbol{\theta}), \quad \text{with} \quad \mathcal{L}^{\text{CL}}(\boldsymbol{\theta}) \triangleq \frac{\lambda}{2} \sum_i S_{k,i}(\theta_i - \bar{\theta}_{k,i}^*)^2 , \tag{5}$$

where $\bar{\theta}_{k,i}^* = \frac{P_{k,i}}{S_{k,i}}$, $S_{k,i} = \sum_{\kappa=1}^k Q_{\kappa,i}$, $P_{k,i} = \sum_{\kappa=1}^k Q_{\kappa,i}\theta_{\kappa,i}^*$, and $Q_{\kappa,i}$ is either $F_{\kappa,i}$ or $\omega_{\kappa,i}$, depending on the method. We name models with EWC and IS augmentations EWC-GAN and IS-GAN, respectively.

---

[1]Zenke et al. (2017) instead consider $\Omega_{1,i} = \frac{\omega_{1,i}}{(\Delta_{1,i})^2+\xi}$, where $\Delta_{1,i} = \theta_{1,i} - \theta_{0,i}$ and $\xi$ is a small number for numerical stability. We however found that the inclusion of $(\Delta_{1,i})^2$ can lead to the loss exploding and then collapsing as the number of tasks increases and so omit it. We also change the hyperparameter $c$ into $\frac{\lambda}{2}$.

**Controlled forgetting**: To provide a mechanism for forgetting earlier non-optimal versions of the discriminator and to keep $\mathcal{L}^{\mathrm{CL}}$ bounded, we add a discount factor $\gamma$: $S_{k,i} = \sum_{\kappa=1}^{k} \gamma^{k-\kappa} Q_{\kappa,i}$ and $P_{k,i} = \sum_{\kappa=1}^{k} \gamma^{k-\kappa} Q_{\kappa,i} \theta_{\kappa,i}^*$. Together, $\alpha$ and $\gamma$ determine how far into the past the discriminator remembers previous generator distributions, and $\lambda$ controls how important memory is relative to the discriminator loss. Note, the terms $S_k$ and $P_k$ can be updated every $\alpha$ steps in an online fashion:

$$S_{k,i} = \gamma S_{k-1,i} + Q_{k,i}, \qquad P_{k,i} = \gamma P_{k-1,i} + Q_{k,i} \theta_{k,i}^* \tag{6}$$

This allows the EWC or IS loss to be applied without necessitating storing either $Q_k$ or $\boldsymbol{\theta}_k^*$ for every task $k$, which would quickly become too costly to be practical. Only a single variable to store a running average is required for each of $S_k$ and $P_k$, making this method space efficient.

Augmenting the discriminator with the continual learning loss, the GAN objective becomes:

$$\min_{\phi} \max_{\theta} \mathcal{L}^{\mathrm{CL}}(\boldsymbol{\theta}, \boldsymbol{\phi}) = \mathcal{L}^{\mathrm{GAN}}(\boldsymbol{\theta}, \boldsymbol{\phi}) - \mathcal{L}^{\mathrm{CL}}(\boldsymbol{\theta}) \tag{7}$$

Note that the training of the generator remains the same; full algorithms are in Appendix A. Here we have shown two methods to mitigate catastrophic forgetting for the original GAN; however, the proposed framework is applicable to almost all of the wide range of GAN setups.

## 4 RELATED WORK

**Continual learning in GANs**   There has been previous work investigating continual learning within the context of GANs. Improved GAN (Salimans et al., 2016) introduced historical averaging, which regularizes the model with a running average of parameters of the most recent iterations. Simulated+Unsupervised training (Shrivastava et al., 2017) proposed replacing half of each minibatch with previous generator samples during training of the discriminator, as a generated sample at any point in time should always be considered fake. However, such an approach necessitates a historical buffer of samples and halves the number of current samples that can be considered. Continual Learning GAN (Seff et al., 2018) applied EWC to GAN, as we have, but used it in the context of the class-conditioned generator that learns classes sequentially, as opposed to all at once, as we propose. Thanh-Tung et al. (2018) independently reached a similar conclusion on catastrophic forgetting in GANs, but focused on gradient penalties and momentum on toy problems.

**Multiple network GANs**   The heart of continual learning is distilling a network's knowledge through time into a single network, a temporal version of the ensemble described in Hinton et al. (2015). There have been several proposed models utilizing multiple generators (Hoang et al., 2018; Ghosh et al., 2018) or multiple discriminators (Durugkar et al., 2017; Neyshabur et al., 2017), while Bayesian GAN (Saatchi & Wilson, 2017) considered distributions on the parameters of both networks, but these all do not consider time as the source of the ensemble. Unrolled GAN (Metz et al., 2017) considered multiple discriminators "unrolled" through time, which is similar to our method, as the continual learning losses also utilize historical instances of discriminators. However, both EWC-GAN and IS-GAN preserve the important parameters for prior discriminator performance, as opposed to requiring backpropagation of generator samples through multiple networks, making them easier to implement and train.

**GAN convergence**   While GAN convergence is not the focus of this paper, convergence does similarly avoid mode collapse, and there are a number of works on the topic (Heusel et al., 2017; Unterthiner et al., 2018; Nagarajan & Kolter, 2017; Mescheder et al., 2017). From the perspective of Heusel et al. (2017), EWC or IS regularization in GAN can be viewed as achieving convergence by slowing the discriminator, but per parameter, as opposed to a slower global learning rate.

## 5 EXPERIMENTS

### 5.1 DISCRIMINATOR CATASTROPHIC FORGETTING

While Figure 1 implies catastrophic forgetting in a GAN discriminator, we can show this concretely. To do so, we first train a DCGAN (Radford et al., 2016) on the MNIST dataset. Since the generator is capable of generating an arbitrary number of samples at any point, we can randomly draw 70000 samples to comprise a new, "fake MNIST" dataset at any time. By doing this at regular intervals, we create datasets $\{\mathcal{D}_1^{\mathrm{gen}}, ..., \mathcal{D}_T^{\mathrm{gen}}\}$ from $p_{\mathrm{gen}}(x)$ at times $1, ..., T$. Samples are shown in Appendix B.

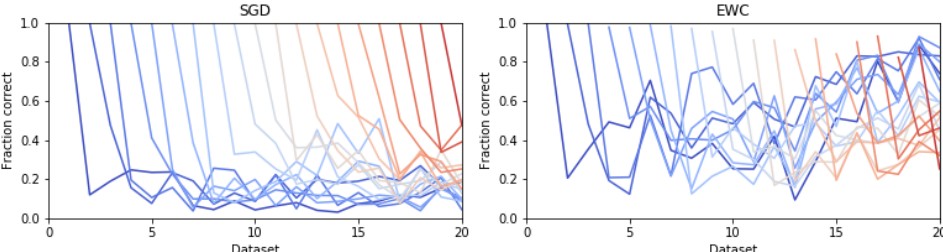

Figure 2: Each line represents the discriminator's test accuracy on the fake GAN datasets. Note the sharp decrease in the discriminator's ability to recognize previous fake samples upon fine-tuning on the next dataset using SGD (left). Forgetting still occurs with EWC (right), but is less severe.

Having previously generated a series of datasets during the training of a DCGAN, we now reinitialize the discriminator and train to convergence on each $\mathcal{D}_t^{\text{gen}}$ in sequence. Importantly, we do *not* include samples from $\mathcal{D}_{<t}^{\text{gen}}$ while fine-tuning on $\mathcal{D}_t^{\text{gen}}$. After fine-tuning on the train split of dataset $\mathcal{D}_t^{\text{gen}}$, the percentage of generated examples correctly identified as fake by the discriminator is evaluated on the test splits of $\mathcal{D}_{\leq t}^{\text{gen}}$, with and without EWC (Figure 2). The catastrophic forgetting effect of the discriminator trained with SGD is clear, with a steep drop-off in discriminating ability on $\mathcal{D}_{t-1}^{\text{gen}}$ after fine-tuning on $\mathcal{D}_t^{\text{gen}}$; this is unsurprising, as $p_{\text{gen}}(x)$ has evolved specifically to deteriorate discriminator performance. While there is still a dropoff with EWC, forgetting is less severe.

While the training outlined above is not what is typical for GAN, we choose this set-up as it closely mirrors the continual learning literature. With recent criticisms of some common continual learning benchmarks as either being too easy or missing the point of continual learning (Farquhar & Gal, 2018), we propose GAN as a new benchmark providing a more realistic setting. From Figure 2, it is clear that while EWC certainly helps, there is still much room to improve with new continual learning methods. However, the merits of GAN as a continual learning benchmark go beyond difficulty. While it is unclear why one would ever use a single model to classify successive random permutations of MNIST (Goodfellow et al., 2013), many real-world settings exist where the data distribution is slowly evolving. For such models, we would like to be able to update the deployed model without forgetting previously learned performance, especially when data collection is expensive and thus done in bulk sometime before deployment. For example, autonomous vehicles (Huval et al., 2015) will eventually encounter unseen car models or obstacles, and automated screening systems at airport checkpoints (Liang et al., 2018) will have to deal with evolving bags, passenger belongings, and threats. In both cases, sustained effectiveness requires a way to appropriately and efficiently update the models for new data, or risk obsolescence leading to dangerous blindspots.

Many machine learning datasets represent singe-time snapshots of the data distribution, and current continual learning benchmarks fail to capture the slow drift of the real-world data. The evolution of GAN synthesized samples represents an opportunity to generate an unlimited number of smoothly evolving datasets for such experiments. We note that while the setup used here is for binary real/fake classification, one could also conceivably use a conditional GAN (Mirza & Osindero, 2014) to generate an evolving multi-class classification dataset. We leave this exploration for future work.

## 5.2  MIXTURE OF EIGHT GAUSSIANS

We show results on a toy dataset consisting of a mixture of eight Gaussians, as in the example in Figure 1. Following the setup of (Metz et al., 2017), the real data are evenly distributed among eight 2-dimensional Gaussian distributions arranged in a circle of radius 2, each with covariance $0.02I$ (see Figure 4). We evaluate our model with Inception Score (ICP) (Salimans et al., 2016), which gives a rough measure of diversity and quality of samples; higher scores imply better performance, with the true data resulting in a score of around 7.870. For this simple dataset, since we know the true data distribution, we also calculate the symmetric Kullback–Leibler divergence (Sym-KL); lower scores mean the generated samples are closer to the true data. We show computation time, measured in numbers of training iterations per second (Iter/s), averaged over the full training of a model on a single Nvidia Titan X (Pascal) GPU. Each model was run 10 times, with the mean and standard deviation of each performance metric at the end of 25K iterations reported in Table 1.

Table 1: Iterations per second, inception score, and symmetric KL divergence comparison on a mixture of eight Gaussians.

| Model | | | | | | |
|---|---|---|---|---|---|---|
| Method | $\alpha$ | $\lambda$ | $\gamma$ | Iter/s $\uparrow$ | ICP $\uparrow$ | Sym-KL $\downarrow$ |
| GAN | - | - | - | $87.59 \pm 1.45$ | $2.835 \pm 2.325$ | $19.55 \pm 3.07$ |
| GAN + $\ell_2$ weight | 1 | 0.01 | 0 | | $5.968 \pm 1.673$ | $15.19 \pm 2.67$ |
| GAN + historical avg. | 1 | 0.01 | 0.995 | | $7.305 \pm 0.158$ | $13.32 \pm 0.88$ |
| GAN + SN | - | - | - | $49.70 \pm 0.13$ | $6.762 \pm 2.024$ | $13.37 \pm 3.86$ |
| GAN + IS | 1000 | 100 | 0.8 | $42.26 \pm 0.35$ | $7.039 \pm 0.294$ | $15.10 \pm 1.51$ |
| GAN + IS | 100 | 10 | 0.98 | $42.29 \pm 0.10$ | $7.500 \pm 0.147$ | $11.85 \pm 0.92$ |
| GAN + IS | 10 | 100 | 0.99 | $41.07 \pm 0.07$ | $7.583 \pm 0.242$ | $11.88 \pm 0.84$ |
| GAN + SN + IS | 10 | 100 | 0.99 | $25.69 \pm 0.09$ | $7.699 \pm 0.048$ | $11.10 \pm 1.18$ |
| GAN + EWC | 1000 | 100 | 0.8 | $82.78 \pm 1.55$ | $7.480 \pm 0.209$ | $13.00 \pm 1.55$ |
| GAN + EWC | 100 | 10 | 0.98 | $80.63 \pm 0.39$ | $7.488 \pm 0.222$ | $12.16 \pm 1.64$ |
| GAN + EWC | 10 | 10 | 0.99 | $73.86 \pm 0.16$ | $7.670 \pm 0.112$ | $11.90 \pm 0.76$ |
| GAN + SN + EWC | 10 | 10 | 0.99 | $44.68 \pm 0.11$ | $7.708 \pm 0.057$ | $11.48 \pm 1.12$ |

The performance of EWC-GAN and IS-GAN were evaluated for a number of hyperparameter settings. We compare our results against a vanilla GAN (Goodfellow et al., 2014), as well as a state-of-the-art GAN with spectral normalization (SN) (Miyato et al., 2018) applied to the discriminator. As spectral normalization augments the discriminator loss in a way different from continual learning, we can combine the two methods; this variant is also shown.

Note that a discounted version of discriminator historical averaging (Salimans et al., 2016) can be recovered from the EWC and IS losses if the task rate $\alpha = 1$ and $Q_{k,i} = 1$ for all $i$ and $k$, a poor approximation to both the Fisher information matrix diagonal and importance measure. If we also set the historical reference term $\bar{\theta}_k^*$ and the discount factor $\gamma$ to zero, then the EWC and IS losses become $\ell_2$ weight regularization. These two special cases are also included for comparison.

We observe that augmenting GAN models with EWC and IS consistently results in generators that better match the true distribution, both qualitatively and quantitatively, for a wide range of hyperparameter settings. EWC-GAN and IS-GAN result in a better ICP and FID than $\ell_2$ weight regularization and discounted historical averaging, showing the value of prioritizing protecting important parameters, rather than all parameters equally. EWC-GAN and IS-GAN also outperform a state-of-the-art method in SN-GAN. In terms of training time, updating the EWC loss requires forward propagating a new minibatch through the discriminator and updating $S$ and $P$, but even if this is done at every step ($\alpha = 1$), the resulting algorithm is only slightly slower than SN-GAN. Moreover, doing so is unnecessary, as higher values of $\alpha$ also provide strong performance for a much smaller time penalty. Combining EWC with SN-GAN leads to even better results, showing that the two methods can complement each other. IS-GAN can also be successfully combined with SN-GAN, but it is slower than EWC-GAN as it requires tracking the trajectory of parameters at each step. Sample generation evolution over time is shown in Figure 4 of Appendix C.

## 5.3 Image generation of CelebA and CIFAR-10

Since EWC-GAN achieves similar performance to IS-GAN but at less computational expense, we focus on the former for experiments on two image datasets, CelebA and CIFAR-10. Our EWC-GAN implementation is straightforward to add to any GAN model, so we augment various popular implementations. Comparisons are made with the TTUR (Heusel et al., 2017) variants[2] of DCGAN (Radford et al., 2016) and WGAN-GP (Gulrajani et al., 2017), as well as an implementation[3] of a spectral normalized (Miyato et al., 2018) DCGAN (SN-DCGAN). Without modifying the learning rate or model architecture, we show results with and without the EWC loss term added to the discriminator for each. Performance is quantified with the Fréchet Inception Distance (FID) (Heusel et al., 2017) for both datasets. Since labels are available for CIFAR-10, we also report ICP for that dataset. Best values are reported in Table 2, with samples in Appendix C. In each model, we see improvement in both FID and ICP from the addition of EWC to the discriminator.

---

[2] https://github.com/bioinf-jku/TTUR
[3] https://github.com/minhnhat93/tf-SNDCGAN

Table 2: Fréchet Inception Distance and Inception Score on CelebA and CIFAR-10

| | CelebA | CIFAR-10 | |
| --- | --- | --- | --- |
| Method | FID $\downarrow$ | FID $\downarrow$ | ICP $\uparrow$ |
| DCGAN | 12.52 | 41.44 | $6.97 \pm 0.05$ |
| DCGAN + EWC | 10.92 | 34.84 | $7.10 \pm 0.05$ |
| WGAN-GP | - | 30.23 | $7.09 \pm 0.06$ |
| WGAN-GP + EWC | - | 29.67 | $7.44 \pm 0.08$ |
| SN-DCGAN | - | 27.21 | $7.43 \pm 0.10$ |
| SN-DCGAN + EWC | - | 25.51 | $7.58 \pm 0.07$ |

Table 3: Test BLEU $\uparrow$ results on MS COCO

| Method | MLE | SeqGAN | RankGAN | GSGAN | LeakGAN | textGAN | EWC | IS |
| --- | --- | --- | --- | --- | --- | --- | --- | --- |
| BLEU-2 | 0.820 | 0.820 | 0.852 | 0.810 | 0.922 | 0.926 | 0.934 | 0.933 |
| BLEU-3 | 0.607 | 0.604 | 0.637 | 0.566 | 0.797 | 0.781 | 0.802 | 0.791 |
| BLEU-4 | 0.389 | 0.361 | 0.389 | 0.335 | 0.602 | 0.567 | 0.594 | 0.578 |
| BLEU-5 | 0.248 | 0.211 | 0.248 | 0.197 | 0.416 | 0.379 | 0.400 | 0.388 |

Table 4: Self BLEU $\downarrow$ results on MS COCO

| Method | MLE | SeqGAN | RankGAN | GSGAN | LeakGAN | textGAN | EWC | IS |
| --- | --- | --- | --- | --- | --- | --- | --- | --- |
| BLEU-2 | 0.754 | 0.807 | 0.822 | 0.785 | 0.912 | 0.843 | 0.854 | 0.853 |
| BLEU-3 | 0.511 | 0.577 | 0.592 | 0.522 | 0.825 | 0.631 | 0.671 | 0.655 |
| BLEU-4 | 0.232 | 0.278 | 0.288 | 0.230 | 0.689 | 0.317 | 0.388 | 0.364 |

## 5.4 TEXT GENERATION OF COCO CAPTIONS

We also consider the text generation on the MS COCO Captions dataset (Chen et al., 2015), with the pre-processing in Guo et al. (2018). Quality of generated sentences is evaluated by BLEU score (Papineni et al., 2002). Since BLEU-$b$ measures the overlap of $b$ consecutive words between the generated sentences and ground-truth references, higher BLEU scores indicate better fluency. Self BLEU uses the generated sentences themselves as references; lower values indicate higher diversity.

We apply EWC and IS to textGAN (Zhang et al., 2017), a recently proposed model for text generation in which the discriminator uses feature matching to stabilize training. This model's results (labeled "EWC" and "IS") are compared to a Maximum Likelihood Estimation (MLE) baseline, as well as several state-of-the-art methods: SeqGAN (Yu et al., 2017), RankGAN (Lin et al., 2017), GSGAN (Jang et al., 2016) and LeakGAN (Guo et al., 2018). Our variants of textGAN outperforms the vanilla textGAN for all BLEU scores (see Table 3), indicating the effectiveness of addressing the forgetting issue for GAN training in text generation. EWC/IS + textGAN also demonstrate a significant improvement compared with other methods, especially on BLEU-2 and 3. Though our variants lag slightly behind LeakGAN on BLEU-4 and 5, their self BLEU scores (Table 4) indicate it generates more diverse sentences. Sample sentence generations can be found in Appendix C.

## 6 CONCLUSION

We observe that the alternating training procedure of GAN models results in a continual learning problem for the discriminator, and training on only the most recent generations leads to consequences unaccounted for by most models. As such, we propose augmenting the GAN training objective with a continual learning regularization term for the discriminator to prevent its parameters from moving too far away from values that were important for recognizing synthesized samples from previous training iterations. Since the original EWC and IS losses were proposed for discrete tasks, we adapt them to the GAN setting. Our implementation is simple to add to almost any variation of GAN learning, and we do so for a number of popular models, showing a gain in ICP and FID for CelebA and CIFAR-10, as well as BLEU scores for COCO Captions. More importantly, we demonstrate that GAN and continual learning, two popular fields studied independently of each other, have the potential to benefit each other, as new continual learning methods stand to benefit GAN training, and GAN generated datasets provide new testing grounds for continual learning.

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

## A  ALGORITHM

We summarize the continual learning GAN implementations in Algorithm 1 and 2.

---

**Algorithm 1** Continual learning GAN with EWC

---

1: **Input**: Training data $\mathcal{D}^{\text{real}}$, latent distribution $p(z)$, hyperparameters of continual learning $\alpha, \gamma, \lambda$, step size $\epsilon$
2: **Output**: $\theta$, $\phi$, and generated samples $\mathcal{D}^{\text{gen}} = \{x_j\}_{j=1}^N$
3: **for** $t = 1, ..., T$ **do**

4:   % `Sample minibatch of size m`
5:   Noise sample: $\{z_j\}_{j=1}^m \sim p(z)$
6:   Data sample: $\{x_j\}_{j=1}^m \sim \mathcal{D}^{\text{real}}$
7:   % `Calculate current discriminator loss`
8:   $\mathcal{L}_\theta = \frac{1}{m} \sum_{j=1}^m [\log D_\theta(x_j)] + \frac{1}{m} \sum_{j=1}^m [\log(1 - D_\theta(G_\phi(z_j)))]$

9:   % `Update history buffer for previous tasks`
10:   **if** $\mod(t, \alpha) = 0$ **then**
11:     % At the end of a task
12:     **for** parameters $\theta_i$ in $\theta$:
13:       $Q_i = \left(\frac{\partial \mathcal{L}_\theta}{\partial \theta_i}\right)^2$ %   $Q$ is the Fisher for EWC
14:       $S_i = \gamma S_i + Q_i$
15:       $P_i = \gamma P_i + Q_i \theta_i^*$
16:       $\bar{\theta}_i^* = \frac{P_i}{S_i}$
17:   **end if**

18:   % `Update discriminator parameter` $\theta$`, adding EWC`
19:   $\bar{\mathcal{L}}_\theta = \mathcal{L}_\theta - \frac{\lambda}{2} \sum_i S_{k,i}(\theta_i - \bar{\theta}_{k,i}^*)$
20:   $\theta_{t+1} \leftarrow \theta_t + \epsilon_t \frac{\partial \bar{\mathcal{L}}_\theta}{\partial \theta_t}$

21:   % `Update generator parameter` $\phi$
22:   $\mathcal{L}_\phi = \frac{1}{m} \sum_{i=1}^m [\log(1 - D_\theta(G_\phi(z_i)))]$
23:   $\phi_{t+1} \leftarrow \phi_t - \epsilon_t \frac{\partial \mathcal{L}_\phi}{\partial \phi_t}$
24: **end for**

---

---

**Algorithm 2** Continual learning GAN with IS

---

1: **Input**: Training data $\mathcal{D}^{\text{real}}$, latent distribution $p(\boldsymbol{z})$, hyperparameters of continual learning $\alpha, \gamma, \lambda$, step size $\epsilon$
2: **Output**: $\boldsymbol{\theta}, \boldsymbol{\phi}$, and generated samples $\mathcal{D}^{\text{gen}} = \{\boldsymbol{x}_j\}_{j=1}^{N}$
3: **for** $t = 1, ..., T$ **do**

4:      % `Sample minibatch of size m`
5:      Noise sample: $\{\boldsymbol{z}_j\}_{j=1}^{m} \sim p(\boldsymbol{z})$
6:      Data sample: $\{\boldsymbol{x}_j\}_{j=1}^{m} \sim \mathcal{D}^{\text{real}}$
7:      % `Calculate current discriminator loss`
8:      $\mathcal{L}_{\boldsymbol{\theta}} = \frac{1}{m} \sum_{j=1}^{m} [\log D_{\boldsymbol{\theta}}(\boldsymbol{x}_j)] + \frac{1}{m} \sum_{j=1}^{m} [\log(1 - D_{\boldsymbol{\theta}}(G_{\boldsymbol{\phi}}(\boldsymbol{z}_j)))]$

9:      % `Update discriminator parameter` $\boldsymbol{\theta}$, `adding IS`
10:     $\bar{\mathcal{L}}_{\boldsymbol{\theta}} = \mathcal{L}_{\boldsymbol{\theta}} - \frac{\lambda}{2} \sum_i S_{k,i}(\theta_i - \bar{\theta}_{k,i}^*)$
11:     $\boldsymbol{g} = \frac{\partial \bar{\mathcal{L}}_{\boldsymbol{\theta}}}{\partial \boldsymbol{\theta}_t}$
12:     $\boldsymbol{\delta} = \epsilon \frac{\partial \bar{\mathcal{L}}_{\boldsymbol{\theta}}}{\partial \boldsymbol{\theta}_t}$
13:     **for** $\theta_i$ in $\boldsymbol{\theta}$ **do**
14:         $\omega_i = \omega_i + g_i \delta_i$
15:     **end for**
16:     $\boldsymbol{\theta}_{t+1} \leftarrow \boldsymbol{\theta}_t + \epsilon_t \frac{\partial \bar{\mathcal{L}}_{\boldsymbol{\theta}}}{\partial \boldsymbol{\theta}_t}$

17:     % `Update history buffer for previous tasks`
18:     **if**    $\mod(t, \alpha) = 0$ **then**
19:         % At the end of a task
20:         **for** parameters $\theta_i$ in $\boldsymbol{\theta}$:
21:             $Q_i = \omega_i$ %    $Q$ is the Importance measure for IS
22:             $S_i = \gamma S_i + Q_i$
23:             $P_i = \gamma P_i + Q_i \theta_i^*$
24:             $\bar{\theta}_i^* = \frac{P_i}{S_i}$
25:             $\boldsymbol{\omega} = \boldsymbol{0}$
26:     **end if**

27:     % `Update generator parameter` $\boldsymbol{\phi}$
28:     $\mathcal{L}_{\boldsymbol{\phi}} = \frac{1}{m} \sum_{i=1}^{m} [\log(1 - D_{\boldsymbol{\theta}}(G_{\boldsymbol{\phi}}(\boldsymbol{z}_i)))]$
29:     $\boldsymbol{\phi}_{t+1} \leftarrow \boldsymbol{\phi}_t - \epsilon_t \frac{\partial \mathcal{L}_{\boldsymbol{\phi}}}{\partial \boldsymbol{\phi}_t}$
30: **end for**

---

## B  GENERATED MNIST DATASETS FOR CONTINUAL LEARNING BENCHMARKING

To produce a smoothly evolving series of datasets for continual learning, we train a DCGAN on MNIST and generate an entire "fake" dataset of 70K samples every 50 training iterations of the DC-GAN generator. We propose learning each of these generated datasets as individual tasks for continual learning. Selected samples are shown in Figure 3 from the datasets $\mathcal{D}_t^{gen}$ for $t \in \{5, 10, 15, 20\}$, each generated from the same 100 samples of $z$ for all $t$. Note that we actually trained a conditional DCGAN, meaning we also have the labels for each generated image. For experiments in Figure 2, we focused on the real versus fake task to demonstrate catastrophic forgetting in a GAN discriminator and thus ignored the labels, but future experiments can incorporate such information.

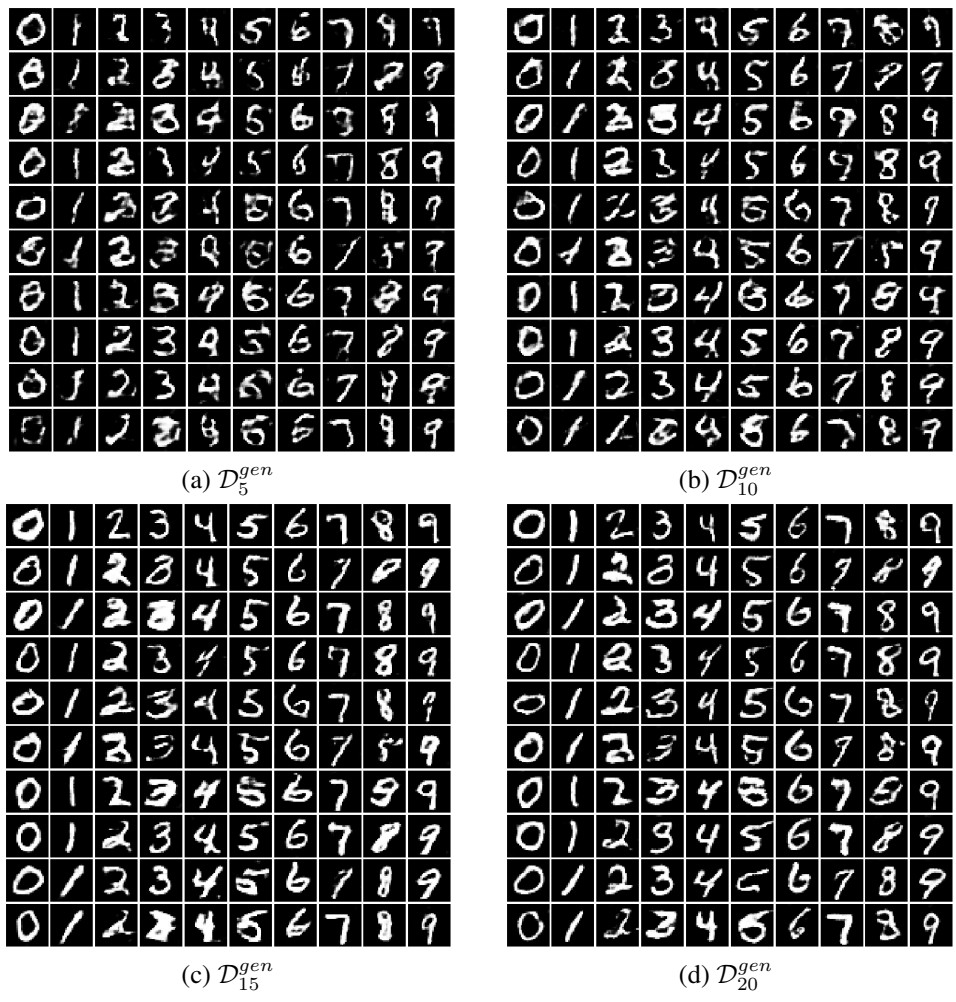

(a) $\mathcal{D}_5^{gen}$      (b) $\mathcal{D}_{10}^{gen}$

(c) $\mathcal{D}_{15}^{gen}$      (d) $\mathcal{D}_{20}^{gen}$

Figure 3: Image samples from generated "fake MNIST" datasets

## C EXAMPLES OF GENERATED SAMPLES

Sample generations are plotted during training at 5000 step intervals in Figure 4. While vanilla GAN occasionally recovers the true distribution, more often than not, the generator collapses and then bounces around. Spectral Normalized GAN converges to the true distribution quickly in most runs, but it mode collapses and exhibits the same behavior as GAN in others. EWC-GAN consistently diffuses to all modes, tending to find the true distribution sooner with lower $\alpha$. We omit IS-GAN, as it performs similarly to EWC-GAN.

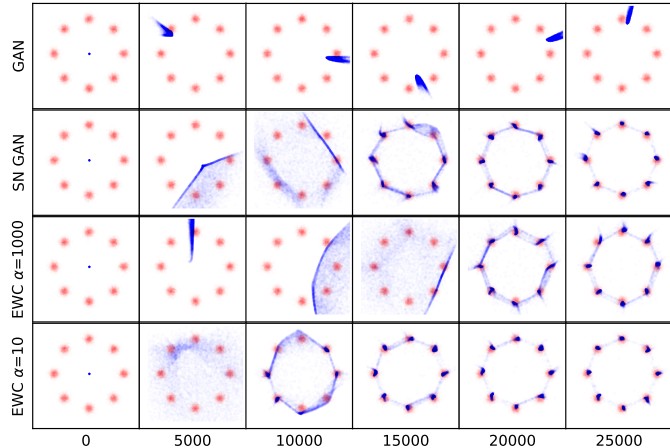

Figure 4: Each row shows the evolution of generator samples at 5000 training step intervals for GAN, SN-GAN, and EWC-GAN for two $\alpha$ values. The proposed EWC-GAN models have hyper-parameters matching the corresponding $\alpha$ in Table 1. Each frame shows 10000 samples drawn from the true eight Gaussians mixture (red) and 10000 generator samples (blue).

We also show the generated image samples for CIFAR 10 and CelebA in Figure 5, and generated text samples for MS COCO Captions in Table 5.

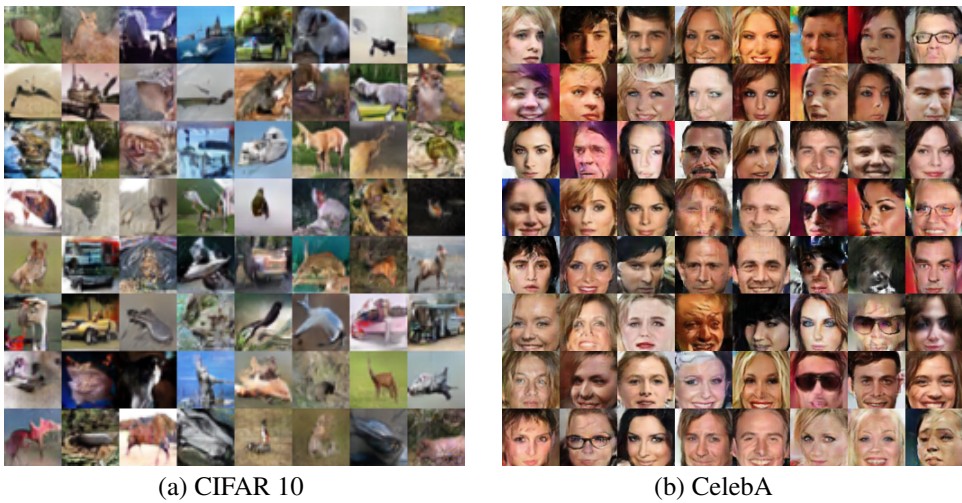

(a) CIFAR 10          (b) CelebA

Figure 5: Generated image samples from random draws of EWC+GANs.

Table 5: Sample sentence generations from EWC + textGAN

a couple of people are standing by some zebras in the background
the view of some benches near a gas station
a brown motorcycle standing next to a red fence
a bath room with a broken tank on the floor
red passenger train parked under a bridge near a river
some snow on the beach that is surrounded by a truck
a cake that has been perform in the background for takeoff
a view of a city street surrounded by trees
two giraffes walking around a field during the day
crowd of people lined up on motorcycles
two yellow sheep with a baby dog in front of other sheep
an intersection sits in front of a crowd of people
a red double decker bus driving down the street corner
an automobile driver stands in the middle of a snowy park
five people at a kitchen setting with a woman
there are some planes at the takeoff station
a passenger airplane flying in the sky over a cloudy sky
three aircraft loaded into an airport with a stop light
there is an animal walking in the water
an older boy with wine glasses in an office
two old jets are in the middle of london
three motorcycles parked in the shade of a crowd
group of yellow school buses parked on an intersection
a person laying on a sidewalk next to a sidewalk talking on a cell phone
a chef is preparing food with a sink and stainless steel appliances

