# OpenReview forum: "Generative Adversarial Network Training is a Continual Learning Problem"
_ICLR.cc/2019/Conference_

### Official Review · AnonReviewer2 · 2018-10-31
**A novel and probably effective plug-and-play regularizer for GANs**

**Rating:** 7
**Confidence:** 4

**Review:**

The authors argue that catastrophic forgetting may cause mode collapse and oscillation, and propose a novel plug-and-play  regularizer that can be applied to a variety of GANs' training process to counter catastrophic forgetting of the discriminator. The regularizer is a clever adaption of EWC and IS into the context of GAN training. With the authors' formulation, this regularizer will account for the discriminator's parameter from all previous "tasks" (snapshots taken at certain iterations) with extra memory budget of only one set of parameters, while assigning higher regularization strengths to parameters learned from recent tasks. Experiments demonstrate such regularizer improves GAN models including DCGAN, SN-DCGAN, WGAN-GP on image generation tasks and textGAN on text generation tasks.

Pros:
The paper is well-written. The formulation of online memory and controlled forgetting are clever, giving rise to the adaption of EWC and IS as a practical regularizer to overcome the problem of catastrophic forgetting in GANs. The experiments also demonstrate the regularizer is superior than historical averaging and SN on the synthetic dataset, and it is able to improve multiple GAN models in both image and text generation tasks.

Cons/Suggestions:
1. Although I can see the method is working, the empirical evidence to support "mode oscillation" is not strong enough for me. I think in order for continual learning to make perfect sense, mode oscillation should be an obvious issue for GANs; otherwise, we probably don't need remembering the history, as the generator is probably evolving towards the right direction even in the vanilla approach. Still, since there have been several papers showing history is important, it should be helpful in some sense. In Figure 1, I cannot tell whether in (d), the generator returned to the previous space (probably refers to (a)). Even the centers of mass of (a) and (d) look different for me. Figure 2 (left) only shows the distribution of generated data is changing as the training proceeds in vanilla GANs, since few of them (some shallow blue lines) have low peaks in previous datasets. If the mode oscillates and the generator returns to previous state, there should at least be another peak along the line, which is missing in curves on later datasets (darker blue ones). (I guess I have understood this figure correctly, but Figure 2 seems horizontally flipped to me. Since you are testing on previous fake datasets, and the accuracy should drop on previous datasets; however, the accuracy drops on later datasets in the figure.)

2. I doubt the authors may not have tried enough sets of hyper parameters for baseline models. In table 1, the variance of GAN, GAN + l2 weight and GAN + SN are significantly higher than the others. I don't think with l2 weight regularizer, the model will be much more unstable than the authors' approach.

3. The authors didn't give the results of their regularizer with LeakGAN on text generation. Currently their model has lower test BLEU than LeakGAN, which indicates lower fluency, but its self BLEU is lower than LeakGAN, which indicates higher diversity. It would be much better if the proposed method can surpass LeakGAN on both metrics.

4. Using inception score on mixture of eight Gaussians may not make much sense, if they are using the ImageNet pre-trained model, since such a model is not trained to fit this distribution. Still, the author has reported symmetric KL.

5. The authors did not specify their inception score on real Celeb-A and CIFAR10 images.


Overall, I tend to accept this paper for its contribution on methods. It would be even better if my concerns could be addressed.

Edit: after seeing the review of Reviewer 3, I find the proposed method seems to be the same as Online EWC and I have downgraded the rating. The authors should address these concerns.

---

> ### Author Response · Authors · 2018-11-14
> **Response to AnonReviewer2**
>
> Thank you for the review. Responses to your comments:
>
> 1.	Apologies if the figures were unclear. With regard to Figure 1, the oscillations in an 8 Gaussians GAN are especially obvious if each time step (each corresponding to a plot like Figure 1 a, b, c, d) are compiled into frames of a video. In such videos, the generator of a vanilla GAN can be seen oscillating through the space in 2D indefinitely, returning to various previous locations repeatedly. For Figure 1, we hoped to demonstrate this compactly with several proximal frames, but given that the space is 2D, the generator at d) doesn’t necessarily return to exactly the same location as a) while oscillating, and it may also visit other modes before returning. Regardless of exact positioning, we can see that the generator is once again producing synthesized samples in d) that it had been disincentivized from producing after a). In Figure 2, the x-axis represents the dataset being trained on, while each line represents performance on a particular dataset as the model is fine-tuned on additional datasets. For example, the darkest blue line represents the model’s test accuracy on D_1, which starts off high when the model is first trained specifically on D_1’s training set, but it drops precipitously once the model is finetuned on D_2.
> 2.	We report the best performances we found from hyperparameter search on our baseline models. The variances for many of the baseline models is higher due to their propensity to fail to converge more often than our methods. This results in at least a few significantly lower ICP/higher Sym-KL values than a converged model, resulting in corresponding higher variances.
> 3.	We agree it would be better if we surpassed LeakGAN on both metrics. However, we would like to emphasize that the proposed method did improve its baseline model textGAN on both metrics, which demonstrates the main point of the utility of continual learning augmentation.
> 4.	Reporting ICP with ImageNet pre-trained features indeed makes little sense. Our ICP is more accurately Inception Score-inspired: we pre-trained a classifier on 8 Gaussians data sample and used that to calculate ICP, much in the way an ImageNet classifier is used for the standard ICP.
> 5.	“Specify inception score on real Celeb-A and CIFAR10 images.” Inception score on real CIFAR10 images is 11.23. Note that Inception score is computed on datasets with labels, which the Celeb-A dataset doesn’t have. Further, we reported FID score on both datasets, which past works have found to be a better metric than Inception score [*].
> [*] GANs Trained by a Two Time-Scale Update Rule Converge to a Local Nash Equilibrium, NIPS 2017
>
> Please see our response to Reviewer 3 for concerns about Online EWC.

---

> > ### Comment · AnonReviewer2 · 2018-11-14
> > **Response**
> >
> > After reading your response, I realized your method has some difference from online EWC and EWC++, and you are using it in a totally different setting from online EWC and EWC++, so I no longer think there is a huge issue about novelty.
> >
> > My concern still remains as to how exactly is such regularizers helpful for GAN training. Basically you are using Figure 1 and Figure 2 to argue that the discriminator forgets previous modes and the generator generates repeated modes.  Although I can see the discriminator forgets to recognize previous modes in Figure 2, there is no clear evidence in it showing that the generator generates previous modes, since if that is true, there should be a peak after the drop. So preventing forgetting might not be necessary.
> >
> > Another concern is about the experiments on text. You should give the result of your regularizer on the best model (LeakGAN).

---

> > > ### Author Response · Authors · 2018-11-15
> > > **Reply to AnonReviewer2**
> > >
> > > Thank you for the quick reply.
> > >
> > > Here’s a video illustrating the oscillations we’re referring to: https://youtu.be/91a2gPWngo8 . We’ve posted it with an anonymized account, to respect ICLR’s double blind policy. We’d like to emphasize that the generator doesn’t need to return to the exact same distribution to demonstrate catastrophic forgetting. Instead, we point out that certain samples--or regions of samples--are returned to repeatedly (especially in the second half of the video), which would be highly disincentivized if the discriminator had remembered them.
> > >
> > > Yes, the generator generating previous modes could result in a peak after the drop. Some possible explanations for Figure 2: i) the rate at which we created the datasets may have been too coarse to capture secondary peaks, ii) compared to the overall length of GAN training, the plotted datasets in Figure 2 represent a relatively narrow slice in time, which may not capture a full period of an oscillation, and iii) the generator may return to only part of a previous distribution (as explained above), in which case the proportion of fake samples resembling previous ones could be small.
> > >
> > > While LeakGAN does have a slightly higher test BLEU score for some window sizes, textGAN shows consistently and significantly better self BLEU scores, indicating higher diversity, so we believe textGAN to actually be the better model, in our experiments.

---

> > > > ### Comment · AnonReviewer2 · 2018-11-15
> > > > **Interesting video**
> > > >
> > > > Thanks for posting the video. I can see the generator returning to previous modes clearly.
> > > >
> > > > Generally I think your paper is decent and novel, despite some small issues. I'm happy to change my ratings. However, I still hope you could provide more evident signs in Figure 1 and Figure 2 to support your assumptions, and demonstrate an overall improvement over LeakGAN or give some analysis of why your regularizer would fail on that in your final version.

---

> > > > > ### Author Response · Authors · 2018-11-29
> > > > > **Updates**
> > > > >
> > > > > We updated Figure 1 with frames from the earlier shared video, to show a more obvious case of the generator oscillations.
> > > > >
> > > > > As requested, we also ran LeakGAN + EWC, and saw marginal improvements for the Test BLEU score:
> > > > >
> > > > >                  LeakGan.    LeakGan + EWC
> > > > > BLEU 2        0.922                      0.928
> > > > > BLEU 3        0.797                      0.786
> > > > > BLEU 4        0.602                      0.603
> > > > > BLEU 5        0.416                      0.442
> > > > >
> > > > > LeakGAN is known to be difficult to train, and we find that it tends to degrade from its pre-trained initialization if run for long. Given the low number of adversarial training steps actually taken, our regularization may not have enough time to provide much benefit. The TextGAN model (as well as the image GANs we considered) has a longer training regimen, in which case catastrophic forgetting is more likely to be a problem. As such, our approach has more opportunity to help TextGAN.

---

> > > > > ### Public Comment · (anonymous) · 2018-12-01
> > > > > **Acknowledgement to the original authors of the idea**
> > > > >
> > > > > I would like to quickly point out that similar experiments were done by [1]. The video is available at
> > > > > https://www.youtube.com/watch?v=eMgU6haZBEc
> > > > >
> > > > > The authors of this paper seem to update the experiment in Figure 1 based on the result of [1] without acknowledging them.
> > > > >
> > > > >
> > > > > [1] On catastrophic forgetting and mode collapse in Generative Adversarial Networks
> > > > > Hoang Thanh-Tung, Truyen Tran, Svetha Venkatesh

---

> > > > > > ### Author Response · Authors · 2018-12-01
> > > > > > **Reply to anonymous**
> > > > > >
> > > > > > We updated Figure 1 with frames from the video (https://youtu.be/91a2gPWngo8) we generated to help answer Reviewer 2’s questions, with the main difference being that we darkened the samples to make them more visible; a sequence similar to Figure 1 can be found starting halfway through the video. We chose to update this figure due to Reviewer 2’s questions on whether the generator was actually returning to previous modes. We believe the new figure makes our point clearer.
> > > > > >
> > > > > > Figure 1 was not taken from the workshop paper cited in [1]. We first observed this behavior ourselves, while running some experiments on 8 Gaussians (which ended up in Table 1). Our experiments were done independently, before [1] was published, and any similarities were unintentional. There was some discussion in an earlier thread with an anonymous commenter in which we pointed out the differences of our work: https://openreview.net/forum?id=SJzuHiA9tQ&noteId=rJgmj-eeJ4&noteId=rke1vMC9c7. After that discussion, we added [1] to our Related Works in our updated version.

---

> > > > > > > ### Public Comment · (anonymous) · 2018-12-02
> > > > > > > **Reply**
> > > > > > >
> > > > > > > Thank you for your answer. In my opinion, section 2 and figure 1 in your paper are very similar to [1]. You should state that the idea was published before. The way you are presenting the idea in your paper makes readers think that the idea is a novel idea.
> > > > > > > [1] was published 3 months before ICLR deadline. I understand that similarities may be unintentional but your paper is making inadequate reference to [1].

---

> > > > > > > ### Comment · AnonReviewer1 · 2018-12-12
> > > > > > > **another ref**
> > > > > > >
> > > > > > > This was also discussed and presented in Fig 2 of unrolled GAN: https://arxiv.org/abs/1611.02163

---

### Official Review · AnonReviewer3 · 2018-11-02
**Not enough**

**Rating:** 3
**Confidence:** 5

**Review:**

This paper connects continual learning and GAN training together, and propose to use standard continual learning schemes (EWC etc) to improve GAN training.

Continual learning for GANs is certainly an important problem to look at. Even though I like the problem, I'm not convinced with the solution provided by the paper.

The paper in it's current form, in terms of technical contributions and experimental analyses presented to support the hypotheses it started with, is not good enough to be accepted in ICLR. My comments:

1) Catastrophic forgetting of discriminator: Interesting view about mode collapse. I have following concerns:
(a) I like the view in which the training is shown as a sequential process. I wonder if we could solve the issue of catastrophic forgetting of discriminator by storing sufficient fake examples from previously generated samples from the generator. Storing previous generations has already been explored, however, just to prove the point that forgetting is an issue, it would be interesting to store enough samples for the mixture of Gaussian setting and analyse.

(b) Why not thinking of catastrophic forgetting of generator? What if we constrain the generator to not-to-forget about previously generated samples by Fisher or something similar? In this case, every new task in the training process will have sufficient fake samples from all the modes.

2) Lack of technical contributions: The approach, in which EWC or IS is being used to regulate the discriminator's parameters, seems to be straightforward. The issue of multiple parameters is being resolved by storing one Fisher/Score vector using moving average type scheme. This, to me, is almost same as Online EWC or EWC++. Both these papers have already addressed this issue and discussed them in great detail. How is this approach different?

3) Not sure what exactly Fig 2 is conveying as D_1^{gen}, \cdots, D_T^{gen} should almost be the same, so training on one and testing forgetting on other doesn't actually say much. I think we can't conclude anything from this figure, at least using MNIST experiments.

Minor:
4) I'm assuming that you call your method EWC/IS GAN (I think it wasn't explicitly mentioned in the paper). Why don't you call Seff et al. 2018 (they used EWC with GAN for the first time) work EWC-GAN? I personally feel that it's important to give acronyms so that it doesn't undermine previous works. Just to clarify, I'm not advocating the work by Seff et al..

5) Please provide citations for mode collapse

Online EWC: Progress & compress: A scalable framework for continual learnin, ICML 2018.
EWC++: Riemannian Walk for Incremental Learning: Understanding Forgetting and Intransigence, ECCV 2018.

---

> ### Author Response · Authors · 2018-11-14
> **Response to AnonReviewer3**
>
> Thank you for the review. Responses to your comments:
>
> 1a. Storing previously generated samples would indeed be an option to combat forgetting, and as we discuss in our Related Works, this has indeed been considered before [1]. However, such an approach necessitates access to previous generations. One could at each time step t either save i) a very large (to prevent memorization) number of samples, or ii) the generator’s weights. Option i) could become very space prohibitive quickly, and ii) even more so. Additionally, injecting old samples limits the number of current samples that can be considered. In our method, we avoid both of these by maintaining memory in our parameters, without requiring a buffer.
> 1b. We indeed considered applying continual learning to the generator, but ultimately decided not to. While it is clear that a discriminator capable of distinguishing any fake from any point in time is desirable, a generator capable of generating any previous fake is not as clearly of use. Additionally, if the discriminator indeed truly retains the ability to recognize any fake from any point in time, then there is no reason for the generator to remember how to fool previous discriminators.
> 2. We approached the problem from the starting point of the characteristics of GAN training. As we outlined in our submission, unlike typical continual learning problems, where there are explicit disjoint tasks (e.g. permutations of MNIST, different Atari games), the definition of a “task” in GAN is less clear cut; this results in a series of challenges that we outline in the bullets of Section 3.2, which we subsequently solve by deriving an in-place update rule. While our ultimate solution may resemble Online EWC and EWC++, we arrived at it independently, to fit the needs of the problem at hand. More specific differences with prior works include our definition of a task rate \alpha, as well as “completing the square” of quadratics to derive our update rule, as opposed to re-centering the posterior on the latest solution.
> 3. Apologies if Figure 2 is not immediately clear. D_1^{gen}, \cdots, D_T^{gen} may appear similar to the human eye, but each D_{t}^{gen} is in fact the result of the generator evolving to make the classifier with weights from time t-1 as poor as possible. Figure 2 illustrates that when the discriminator learns a new generator distribution, catastrophic forgetting results in the severe degradation of the discriminator’s ability to recognize previous generator distributions. This process occurs repeatedly during the training process of a GAN. See our reply to reviewer 1 for additional comments.
> 4. We started referring to our model as EWC-GAN and IS-GAN early during development (before becoming aware of Seff et. al. 2018). The name stuck during our discussions, and since our work isn’t directly comparable with Seff et. al 2018, we didn’t notice the name collision. Would GAN+EWC/GAN+IS be sufficiently different, or would you still consider that too close?
> 5. Noted. They’ll be in our next draft. Among many papers discussing “Mode collapse”, we would like mention a few:
> Mode regularized generative adversarial networks. ICLR 2017.
> Improved techniques for training GANs. NIPS 2016
> VEEGAN: Reducing Mode Collapse in GANs using Implicit Variational Learning, NIPS 2017
>
> [1] Learning from Simulated and Unsupervised Images through Adversarial Training
> Ashish Shrivastava, Tomas Pfister, Oncel Tuzel, Josh Susskind, Wenda Wang, Russ Webb

---

> > ### Comment · AnonReviewer3 · 2018-12-03
> > **Response**
> >
> > 1a. I don't think one needs to store very large number of samples. there are experiments in which even 1-2% of previous samples (MNIST) are enough for substantial improvement in the performance. a toy experiment would have strengthened the paper and made it conceptually more clear.
> >
> > 1b. capable of generating previous fakes is desirable if previous fakes belong to the past modes which the generator couldn't generate from at present. Again, without showing how preventing forgetting of generator isn't useful, one can't justify only focusing on discriminator. lack of such experiments and intuitions make the paper incomplete.
> >
> > 2. Previous points, even though extremely important for the completeness of the paper, could be considered as a part of ablation study. However, one of my main criticisms is novelty. EWC++ and Online EWC were published in ECCV 2018 and ICML 2018. Saying that the authors reached similar formulation independently isn't enough as out of three main contributions of this paper -- (a) Def of task; (b) online memory; and (c) controlled forgetting -- (b) and (c) were discussed and addressed in EWC++ and online EWC.
> >
> > Based on my arguments given above, my review of this paper remains the same. This paper is good enough for a workshop in its current form, however, I strongly believe that the paper isn't novel and thorough enough to be published as a full article in ICLR.

---

### Official Review · AnonReviewer1 · 2018-11-05
**Nice extensions of continual learning techniques, but not sure about GANs as a benchmark**

**Rating:** 5
**Confidence:** 4

**Review:**

Summary:
This paper proposes the use of GANs as a realistic benchmark for continual learning, and shows how continual learning techniques applied to the discriminator can alleviate mode collapse. Existing continual learning approaches for discrete task structure (EWC and IS) are adapted to the continually shifting domain of GANs, and evaluated on a toy mixture of Gaussians, CelebA and CIFAR-10 image generation as well as textGANs.

This is a clearly written paper that nicely addresses some of the challenges of bridging the toy problems of continual learning with a real world problem of GAN training. The experiments and ablations are thorough, but the empirical gains in terms of improving GAN metrics are relatively minor. It's also not obvious that GAN training is really a continual learning problem, as every time the generator distribution shifts, the discriminator has to shift as well. Thus progress on stabilizing and improving GANs might not transfer back to domains that truly represent continual learning where the goal is to build a single network that perform well at all points in time. In terms of more realistic benchmarks for continual learning, I believe a controlled synthetic dataset would be more practical than the sequence of GAN checkpoints proposed here.

Strengths:
+ Clearly written, with good background discussion of continual learning approaches and challenges.
+ Interesting adaptation of EWC and IS to the continual setting with task rates, online memory (sum of quadratics is quadratic), and controlled forgetting with a time decay.
+ Thorough experiments and ablations on toy tasks, CelebA and CIFAR-10, and textGANs. Nicely includes error bars and compares computation time for each approach.

Weaknesses:
- The paper could benefit from more discussion on the goals of continual learning, and what is wrong with existing toy benchmarks. Why not come up with a tractable toy problem that addresses these difficulties directly?
- I remain unconvinced that GANs as a good benchmark for continual learning. For example, it has been argued that many of the problems with GANs arise from dynamics of minimax optimization difficulties, and there are many recent approaches that were not compared to that focus on this optimization aspect of GAN training (Metz et al., Roth et al., Mescheder et al.). How would you relate these theoretical ideas to continual learning?
- Most the experimental improvements are incremental. How did you choose or tune hyperparameters of your approach?

---

> ### Author Response · Authors · 2018-11-14
> **Response to AnonReviewer1**
>
> Thank you for the review. Responses to your comments:
>
> - GAN as a continual learning problem: Our argument comes from the rationale that the ideal discriminator should be able to identify any fake sample, produced from any past generator (G_{\theta_1}, …, G_{\theta_t}) from any point in time, not just the current instantiation of the generator (G_{\theta_t}). If this isn’t the case, because of catastrophic forgetting and the way training samples are presented (only from G_{\theta_t}), the discriminator will forget past generator samples in favor of current ones. As a result, the generator has the option of simply shifting to one of the previous instantiations (g_{\theta_<t}), as can be seen Figure 1. This can result in oscillations in the training process, particularly in mode-collapsed generators. Please see https://youtu.be/91a2gPWngo8 as an example (we've posted this video with an anonymized account, to respect ICLR's double submission policy).
>
> - Continual learning benchmarks: The primary concern with current continual learning benchmarks is that many choose to focus on learning a sequence of classification tasks, often with each task bearing little resemblance to the others. [1] points out that this is actually an “unrealistic best case scenario,” because the forgetting effect is actually minimized, due to the completely unrelated tasks. Our opinion is that in real world applications, continual learning is more likely to be of utility when used for a model encountering a slowly evolving environment, as is likely to happen in the real world. This kind of setting isn’t commonly considered in the continual learning literature, partly because most of the common machine learning datasets are static, collected from a distribution at a single snapshot in time. We are unaware of any common datasets that exhibit a coherent time evolution, which if they existed could be used to test continual learning in an evolving setting. Because of GAN’s evolving nature and impressive generative capabilities, we’re proposing GANs as a way to synthetically create such datasets.
>
> - Relation to GAN optimization literature: We are aware of the many works on GAN optimization, but we consider our perspective to be orthogonal to these works. Even with these various techniques to stabilize the minimax, the discriminator is still presented with the task of learning a changing distribution, and as such, GAN training remains a continual learning problem.
>
> - Hyperparameters: We used the 8 Gaussians toy setting to get a feel for how the various hyperparameters interacted with each other. For the real datasets, we used \alpha=100, \gamma=0.99, and \lambda=1e-3, keeping the rest of the hyperparameters the same as the baselines (pre-augmentation) that we compared against.
>
> [1] Sebastian Farquhar and Yarin Gal. Towards Robust Evaluations of Continual Learning. arXiv preprint, 2018.

---

### Public Comment · (anonymous) · 2018-10-09
**Incremental work**

This paper combines several published papers without citing them.
1. Formulating GAN learning as a continual learning problem has been discussed in:
On catastrophic forgetting and mode collapse in Generative Adversarial Networks
Hoang Thanh-Tung et al. ICML 2018 workshop on Theoretical Foundation and Applications of Deep Generative Models. https://arxiv.org/abs/1807.04015
The formulation by Thanh-Tung et al. is very similar to this paper.
2. Using continual learning techniques to help GANs has been studied in
Ari Seff, Alex Beatson, Daniel Suo, and Han Liu. Continual Learning in Generative Adversarial
Nets. arXiv preprint, 2018.
and Thanh-Tung et al.  This paper makes a small incremental contribution over these two papers.

Some other works on continual learning in DGMs are:
1. Variational Continual Learning
Cuong V. Nguyen, Yingzhen Li, Thang D. Bui, Richard E. Turner.  https://arxiv.org/abs/1710.10628

---

> ### Author Response · Authors · 2018-10-10
> **Response to suggested Related Work**
>
> Thanks for pointing out the related work.
>
> It is interesting to know the workshop paper [1], which provides a similar perspective on the catastrophic forgetting issues in GAN training dynamics. We’ll cite it in our updated version. However, we emphasize that (a) Our work was done independently of (and likely concurrently with) [1]. (b) The proposed solutions are different: our algorithms are inspired and built upon continual learning methods, while [1] focuses on gradient penalties and momentum. (c) Experiments in [1] primarily focus on a toy example; while we use the same toy example to illustrate our point, we also conducted extensive experiments on real datasets (CIFAR10, CelebA, MS COCO).
>
> [2] is actually already cited and discussed in the first paragraph of our Related Work. We want to emphasize here that while the title may seem similar, [2] addresses a completely different problem. [2] doesn’t identify that the evolution of a GAN generator during training results in a changing task for the discriminator, as we have, and as such, [2] doesn’t attempt to prevent the discriminator catastrophic forgetting inherent to the GAN training regimen. Rather, [2] is a conditional GAN that seeks to learn each class of a dataset, one at a time. This is more akin to replacing the classical continual learning benchmarks of sequential classification with sequential conditional generation, rather than addressing a problem inherent to GAN as we have.
>
> Continual learning (CL) is becoming a popular topic, and there are many recently proposed algorithms. We would have liked to discuss/cite many of the other recent CL algorithms (e.g. [3]), but given space constraints, kept our focus narrow. Note that we mention in our submission that our perspective enables any CL-based algorithm (e.g. [3]) to be adapted to stabilize GAN training. We’ll consider citing more CL algorithms in the final submission.
>
> [1] On catastrophic forgetting and mode collapse in Generative Adversarial Networks
> [2] Continual Learning in Generative Adversarial Nets
> [3] Variational Continual Learning

---

> > ### Public Comment · (anonymous) · 2018-10-10
> > **Response**
> >
> > Thank you for your detailed reply. In my opinion [1] has a more thorough analysis of the catastrophic forgetting problem. [1] also mentioned Synaptic Intelligence and EWC as possible solutions for catastrophic forgetting in their (your) setting. Actually, a simplified version of EWC which simply performs weight averaging over timesteps was used in ProgressiveGAN [5] and latter in [4] and many other works (I haven't checked carefully but I think BigGAN might use a similar technique). My guess is that the authors of [1] are aware of this so they didn't extend their research further or they are working on a more advanced algorithm.
> >
> > [4] Which Training Methods for GANs do actually Converge?
> > Lars Mescheder, Andreas Geiger, Sebastian Nowozin
> > [5] Progressive Growing of GANs for Improved Quality, Stability, and Variation
> > Tero Karras, Timo Aila, Samuli Laine, Jaakko Lehtinen

---

> > > ### Author Response · Authors · 2018-10-10
> > > **Author response**
> > >
> > > [1] does mention EWC and Synaptic Intelligence in their Future Works, but only briefly discusses a naïve approach, which they acknowledge as impractical. We separately arrived at this conclusion (and outline it in our submission), and therefore proposed several solutions in Section 3.2 to make it amenable to GAN. With our proposed methods, we were able to achieve better performance on real datasets, which [1] did not attempt.
> > >
> > > Additionally, we’d like to point out that while we may have independently arrived at a similar perspective to [1] with respect to the catastrophic forgetting problem in GAN, improving GAN by mitigating catastrophic forgetting is not our only contribution. In Section 5.1, we chose to also analyze the extent to which catastrophic forgetting occurs in a discriminator, trained as in the common continual learning benchmarks. We do this because we believe that GAN-generated datasets represent an opportunity to provide more interesting experimental settings for continual learning algorithms. As explained in our paper, the common sequential classification benchmarks in the continual learning literature do not accurately reflect the more likely real-world setting of a slowly evolving data distribution. Such datasets are hard to find, and we propose that GAN can fill this void. As such, continual learning can also benefit from GAN, as well as vice versa. [1] doesn’t discuss this topic.
> > >
> > > We agree that weight averaging is a crude approximation of EWC. That connection was made explicit in Section 5.2 of our submission, and a direct comparison showed the benefits of our approach. BigGAN indeed uses a moving average for its weights, as in [5], but given that BigGAN is also an ICLR 2019 submission (and one that was posted on arXiv after the ICLR deadline), we couldn’t have mentioned it in our submission.

---

> > > > ### Public Comment · (anonymous) · 2018-10-11
> > > > **Some additional comments**
> > > >
> > > > Thank you for your insightful reply. I just want to point out that your statement: "While GAN convergence is not the focus of this paper, convergence does similarly avoid mode collapse" is not necessarily true.
> > > > Convergence is not the same as less mode collapse (or generalization). A counterexample is shown (interestingly, on the same 8 Guassians dataset with very similar gradient plot and stuff) in this submission:
> > > > [6] https://openreview.net/forum?id=ByxPYjC5KQ
> > > >
> > > > GAN could converge to collapsed equilibrium where the optimal response is to stay at the current state. [7] theoretically showed that equilibrium is achievable even when the generator simply remembers a small number of datapoints. Theoretical result is demonstrated with a clear example in [6]. [6] further propose a method for improving both convergence and generalization of GAN. It might be an interesting read for you as it was for me :).
> > > >
> > > > A number of methods for preventing catastrophic forgetting have been proposed in GAN literature, although the authors didn't make it explicit. Another example is experience replay where samples from previous iterations are reintroduced to the discriminator. From the angle of [6], this could be understood as a method for improving the generalization of the discriminator, making it less prone to overfitting to the current dataset.
> > > >
> > > > [7] Generalization and Equilibrium in Generative Adversarial Nets (GANs)
> > > > Sanjeev Arora, Rong Ge, Yingyu Liang, Tengyu Ma, Yi Zhang

---

> > > > > ### Author Response · Authors · 2018-10-14
> > > > > **Reply to additional comments**
> > > > >
> > > > > Thanks for the reading suggestion, and apologies for the vague wording. We were using “convergence” to mean “convergence to the true distribution.”
> > > > >
> > > > > Yes, we agree that experience replay is a way of avoiding catastrophic forgetting of previous generator distributions; to acknowledged this, we cited [8] in our Related Works as an example. As we mention in our submission though, the disadvantage of such an approach is that it requires saving a historical buffer of previous generations, which can get large if a representative number of samples is saved at regular intervals. Saving the generator itself during training is even more prohibitive. Furthermore, such an approach uses up half the discriminator’s training minibatch with stale images, while a continual learning approach allows for training on minibatches consisting of solely the newest generations.
> > > > >
> > > > > [8] Learning from Simulated and Unsupervised Images through Adversarial Training
> > > > > Ashish Shrivastava, Tomas Pfister, Oncel Tuzel, Josh Susskind, Wenda Wang, Russ Webb

---

### Meta-Review · Area_Chair1 · 2018-12-15

**Confidence:** 5
**Recommendation:** Reject

**Metareview:**

This paper studies training of the generative adversarial networks (GANs), specifically the discriminator, as a continual learning problem, where the discriminator does not forget previously generated samples. This model can be potentially used for improving GANs training and for generating synthetic datasets for evaluating continual learning methods. All the reviewers and AC agree that showing how continual learning techniques applied to the discriminator can alleviate mode collapse in GANs training is an important direction to study.

There is reviewer disagreement on this paper. AC can confirm that all three reviewers have read the author responses and have contributed to the final discussion.

While acknowledging that continual learning setting is potentially useful, the reviewers have raised several important concerns: (1) low technical novelty in light of EWC++ and online EWC methods (R1 and R3) -- methodological and empirical comparison to these baselines is required to assess the difference and benefits of the proposed approach; the authors response to these concerns (and also R2’s comments in the discussion) were insufficient to assess the scope of the contribution. (2) More diverse/convincing empirical findings would strengthen the evaluation (e.g. assessing whether or not generator could help to overcome forgetting; showing that memory replay strategy by storing sufficient fake examples from previously generated samples cannot prevent mode collapse in GANs training – see the R3’s comment; showing the benefits of the generated samples for evaluating continual learning methods). (3) R1 left unconvinced that GAN training can be improved via continual learning training, as the relation between the proposed view and the minimax optimization difficulties in GANs is not addressed – see R1’s comment about this. The authors briefly discussed in their response to the review that the proposed approach is orthogonal to these works. However, a better (possibly theoretical) analysis of GANs training and continual learning would indeed help to evaluate the scope of the contribution of this work.

Regarding the available datasets that exhibit a coherent time evolution -- see the Continuous Manifold Based Adaptation for Evolving Visual Domains by Hoffman et al, CVPR 2014.

Among (1)-(3):  (2) and (3) did not have a substantial impact on the decision, but would be helpful to address in a subsequent revision. However, (1) makes it very difficult to assess the benefits of the proposed approach, and was viewed by AC as a critical issue.

AC suggests that in this current state the paper can be considered for a workshop and recommend to prepare a major revision before resubmitting it for the second round of reviews.